# Assessing SARS-CoV-2 Infection Rate among Romanian Dental Practitioners

**DOI:** 10.3390/ijerph18094897

**Published:** 2021-05-04

**Authors:** Ondine Lucaciu, Antonia Boca, Anca Stefania Mesaros, Nausica Petrescu, Ovidiu Aghiorghiesei, Ioana Codruta Mirica, Ioan Hosu, Gabriel Armencea, Simion Bran, Cristian Mihail Dinu

**Affiliations:** 1Department of Oral Health, University of Medicine and Pharmacy “Iuliu Hatieganu”, 400012 Cluj-Napoca, Romania; ondineluc@yahoo.com (O.L.); nausica_petrescu@yahoo.com (N.P.); ovidiu.agh@gmail.com (O.A.); mirica_codruta@yahoo.com (I.C.M.); 2Department of Oral Rehabilitation, University of Medicine and Pharmacy “Iuliu Hatieganu”, 400012 Cluj-Napoca, Romania; 3Department of Dental Propaedeutics, University of Medicine and Pharmacy “Iuliu Hatieganu”, 400006 Cluj-Napoca, Romania; mesaros.anca@umfcluj.ro; 4Department of Communication, Public Relations and Marketing, Faculty of Political, Administrative and Communication Sciences, Babes Bolyai University, 400132 Cluj-Napoca, Romania; ioan.hosu@ubbcluj.ro; 5Department of Maxillofacial Surgery and Implantology, University of Medicine and Pharmacy “Iuliu Hatieganu”, 400029 Cluj-Napoca, Romania; garmencea@gmail.com (G.A.); dr_brans@yahoo.com (S.B.); dinu_christian@yahoo.com (C.M.D.)

**Keywords:** COVID-19, pandemic, dentistry, dental practice

## Abstract

Due to the impact of the Covid-19 pandemic on dental treatments, the present research aimed to assess the infection rate among dental practitioners from Romania and to analyze the economic impact of Covid-19 on dental offices. We designed a web-based survey distributed to dental practitioners from Romania. The survey included questions that assessed demographic data from the dentists who completed the questionnaire, along with economic aspects and epidemiological aspects related to the impact of the COVID-19 pandemic on dental practitioners. Five hundred and seven dentists completed the survey. Three-quarters of the assessed dental offices reported a decrease in the income and patient volume compared to 2019. More than half of the patients visiting the dental office paid more attention to the risk of infection and to prevention methods. Most dental offices implemented specific protective equipment for doctors. Three thousand seven hundred thirty-five dental practitioners were working in the 507 assessed dental offices, and among them, 238 COVID-19 cases of contamination were recorded. High contamination rates were registered in October (48, 20.1%), November (66, 27.7%), and December 2020 (52, 21.85%). Contamination mostly took place at home (114 cases, 47.8%) or resulted from event attendance. This study underlines an overall greater level of safety and an increased patient addressability in dental offices.

## 1. Introduction

Since the outbreak of the severe acute respiratory syndrome coronavirus 2 (SARS-CoV-2), more than 118,638,190 cases and 2,632,074 deaths have been recorded worldwide [1]. In Romania, 913,143 cases and 22,579 deaths were recorded until 24 March 2021 [1]. The surge in the number of infected people in Romania and across the globe had an important social and economic impact and hampered health care delivery.

The main mode of transmission of SARS-CoV-2, the causative agent for the infectious disease COVID-19, is airborne via person-to-person contact. Experts also support the asymptomatic and pre-symptomatic transmission of the virus [2,3]. During breathing, speech, coughing, or sneezing, virus-containing droplets (5–12 micrometers) and aerosols (<5 mm) are released in the environment. Once in the air, virus-containing droplets and aerosols can be inhaled, or they can contaminate surfaces [4].

In dental offices, aerosol-generating dental procedures (AGDPs) are amongst the most common types of treatment; thus, concerns about the risk of transmission from patient-to-practitioner and patient-to-patient arose [5,6,7,8]. The American Dental Association (ADA) and the Centers for Disease Control and Prevention (CDC) released in April and May 2020 the guidelines on infection control protocols and changes to the practice and office environments [9,10,11].

In Romania, a state of emergency was declared on 18 March 2020, provided by Military Ordinance [12], which consisted of several restrictions such as the closure of schools, universities, and research institutes, cancellation of public events, closures of churches, and cancellation of religious services and events, isolation and quarantine of the infected persons, and social isolation of the general population [13]. As a result, dental offices were also officially closed, and only a few selected dental offices in each region remained open for emergency care.

Two months later, on 18 May 2020, Romania declared a state of alert, and dental offices could resume routine care, although new recommendations for dental offices were introduced [14]. Dental treatments could be performed only with appropriate protective equipment for caregivers, and a set of specific protective measures for patients had to be adopted (phone triage [15], COVID-19 patient screening questionnaire, avoidance of dental scaling, reduction of the number of patients/day, disinfecting mats, hand disinfection use for patients, ventilation between patients, limitation of individuals accompanying patients, use of aerosols evacuators, and separate staff-patient paths).

In Romania, patients’ fear of exposure to SARS-CoV-2 has led to a decreased addressability in dental offices during the state of emergency period, even in case of emergency treatments [16]. This trend was also noted in the months following the state of emergency. However, dental offices implemented new protocols for SARS-CoV-2 transmission-prevention, and it could be argued, in this respect, that the COVID-19 pandemic produced a series of major changes that strongly impacted dental treatment delivery, oral health, and quality of life of patients.

Therefore, this study was aimed to determine the contamination rate among Romanian dental practitioners and highlights the importance of clearly assessing whether infections were the result of community transmission or whether they were associated with oral health care delivery. Therefore, the compliance of dental practitioners with recommendations for preventing SARS-CoV-2 transmission is an important aspect that needs further evaluation. A better understanding of the contamination rate in dental offices and higher compliance with the latest recommendations for the prevention of SARS-CoV-2 transmission is likely to increase the addressability of patients in dental offices. However, decreased addressability in dental offices, also associated with fear of SARS-CoV-2 contamination, could negatively impact the oral health status of the patient.

In light of these challenges, the main purpose of the present research is to assess the infection rate among dental practitioners from Romania and to analyze the economic impact of COVID-19 on dental offices.

## 2. Materials and Methods

Study design: a cross-sectional study was conducted using a web-based survey distributed via Google Forms. The survey was carried out from 26 December 2020 to 1 March 2021. Dental practitioners from both the private and public sectors were invited to complete the survey. The survey was distributed via the internet (e-mail or social media) on different dentistry forums and discussion rooms. Participants were voluntarily involved in this study, and informed consent was obtained from each of them. All participants were informed that no personal data was collected and agreed to complete the survey anonymously.

The study received the “Iuliu Haţieganu” University of Medicine and Pharmacy Ethical Committee approval (383/23.12.2020).

A preliminary questionnaire consisting of 32 questions (Appendix A) was designed by 4 dentists, one psychologist, one expert in social sciences, and a statistician. The validity and content of the questionnaire were assessed. The questionnaire was pilot tested with 15 dentists.

The survey was completed by only one dental practitioner from each dental clinic. The questionnaire consisted of 3 sections: demographic and professional data assessment of the dental practitioners who completed the questionnaire, economic aspects, and, respectively, epidemiological aspects related to SARS-CoV-2 infection among dental practitioners (Appendix A).

Results were stored in a Microsoft Excel database. Members of the present research group reviewed data for accuracy. Descriptive statistics were performed using an online Statistical Package for the Social Sciences Statistic: Descriptive Statistics, Social Science Statistics, Jeremy Stangroom, Retrieved from https://www.socscistatistics.com/descriptive/default2.aspx. (accessed on 12 March 2021). Correlation between evaluated parameters was determined using the Chi-square test, and a *p*-value of <0.05 was considered statistically significant (Chi-Square Test Calculator. Social Science Statistics, Jeremy Stangroom, https://www.socscistatistics/test/chisquare2/default2.aspx) (accessed on 16 March 2021). Logistic regression was performed using Quest Graph™ Logistic Regression (Logit) Calculator (AAT Bioquest, Inc, Sunny Vale, CA, USA). Retrieved from https://www.aatbio.com/tools/logistic-regression-logit-calculator (accessed on 27 April 2021).

## 3. Results

### 3.1. Demographic and Professional Data of the Dental Practitioners who Completed the Survey

A total number of 507 dentists completed the survey. Each doctor completed the information per dental clinic. Three hundred and thirty-eight (66.70%) of the persons who completed the questionnaire were managers, and the rest, 169 (33.34%), were employees. The total number of doctors and auxiliary staff working in the assessed dental offices was 3735 out of which 1811 were doctors. Of the practitioners who completed the survey, 382 (75.3%) were female, and 125 (24.7%) were male. The age groups 30–39 and 40–49 years were the most involved in the completion of the survey. Most of the respondents were general dentists working in a private setting. Four hundred and fifty-three (89.34%) of the doctors who completed the survey were working in private settings, 10 (1.9%) in public dental settings, and 44 (8.7%) worked in both private and public settings. There were 467 (92.1%) practitioners who provided data for dental offices from urban areas and 40 (7.9%) respondents who reported for rural areas.

### 3.2. Economic Aspects Correlated with the COVID-19 Pandemic

During the state of emergency in Romania from 18 March to 18 May 2020, only 39 (7.7%) out of the assessed dental offices provided dental treatments. Following 18 May 2020, when all of the assessed dental offices reopened, the estimated number of emergency dental treatments was roughly the same, 270 (53.3%), as before the onset of the pandemic. According to our survey, 272 (53.5%) of the practices met the same demand concerning the range of services as before the pandemic, while 213 (42%) were confronted with a decrease in the variety of dental treatments demanded by the patients. During the state of emergency, the temporary lockdown of dental clinics for two months, as well the lower demand for elective dental services, impacted financial incomes for 2020. Thus, 382 (75.2%) of the assessed dental offices reported a decrease in income. Thus 102 (26.7%) showed a 0–20% reduction in income, whereas 168 (44% of the total amount) were characterized by a 20–40% reduction (Figure 1). It is important to mention that despite income decreases, the majority of the assessed dental offices maintained the same team size (Table 1).

In relation to the dental offices facing revenue decline due to the pandemic, our study indicates that 138 (27.2%) raised dental treatment fees, and 43 (8.5%) introduced a COVID-19 tax. There is no statistically significant difference (*p* > 0.5) regarding revenue decline between dental offices that raised the fees for dental treatments and/or introduced a COVID-19 tax, compared with other dental offices that did not introduce COVID-19 tax or increased dental treatment prices.

The assessed dental offices stated that most of the patients visiting between May 2020 and March 2021 paid more attention to the risk of infection, as noted by 289 (57%) participants, and were believed to be more interested in the prevention methods adopted by the dental office in 281 (55.4%) cases (Figure 2). The dental offices enrolled in the survey declared that 0–10% of patients canceled their appointment after being testing COVID-19 positive.

More than 50% of the assessed dental offices declared that patient volume was lower between May 2020 and December 2020. Starting January 2021, a significant increase in the number of patients was recorded (Table 2). In contrast, the most noticeable decrease was recorded in May 2020 compared to the same month of 2019, but this dip in patient volume could be attributed to the two-week closure of clinics imposed by authorities during the lockdown (18 March–18 May 2020).

Taking into consideration the new protocols for COVID-19 prevention, most of the dental offices, 496 (97.83%), implemented specific/modified protective equipment for doctors compared to the pre-COVID-19 period, used UV lamps 482 (91.12%), implemented new protocols for patients: proper time gaps between patients 455 (89.74%), patient temperature check 485 (95.66%), protection equipment for patients 373 (73.57%) and phone triage 416 (82.05%) (Figure 3). In most cases, treatment fees were not increased 323 (63.1%), and in 453 (89.3) cases, no COVID-19 tax was charged.

In the case of dental offices that provided specific/modified equipment for staff and patients, UV lamps, and proper time gaps between patients, 13 (2.94%) infections were reported. Regarding the effectiveness of these prevention measures, the following was observed:Evidence suggests that using a nebulizer or a UV lamp and patient phone triage was not more efficient than temperature checks alone in the dental office;Testing the staff was more efficient than testing the patients (χ^2^ = 8.19, *p* = 0.0042);Enhanced personal protective equipment for dental staff was more efficient than the use of gloves, robe, and cap for patients (χ^2^ = 121.7, *p* = 0.0001);Patient phone triage was more efficient than equipment (gloves, robe, and cap) for patients (χ^2^ = 16.28, *p* = 0.00055);Allowing proper time gaps between patients was more efficient than using enhanced protective equipment for doctors (χ^2^ = 15.53, *p* = 0.00081) or patients (χ^2^ = 16.28, *p* = 0.00055);The difference in the infections between offices who respect a rigorous interval between patients and those who do not respect it is in favor of the former (fewer infections) χ^2^ = 16.28, *p* = 0.00042;Dental offices that performed mandatory testing for staff reported a higher number of infections than those that did not implement such a requirement, and the difference was significant χ^2^ = 6.12, *p* = 0.01336.

Among the additional measures enacted by some of the dental offices enrolled in our survey, we could include: COVID-19 questionnaire forms for patients regarding possible positive contacts (16), avoidance of dental scaling (3), reduction of the number of patients/day (8), disinfecting door-mats (8), hand disinfection for patients (11), ventilation between patients, limitation of individuals accompanying patients, use of aerosols evacuators, separate staff-patient paths, and systematic disinfection of waiting room areas or frequently touched surfaces and objects such as toilets or door handles.

### 3.3. Epidemiological Aspects Related to SARS-CoV-2 Infection among Dental Practitioners

A total of 3735 dental practitioners were working in the 507 assessed dental offices, and 283 COVID-19 cases of contamination were recorded, of which 61 (25.63%) were male and 177 (74.37%) were female. The most affected groups were aged 30–39 (97, 40.76%) and 40–49 (68, 28.57%). All these cases were reported by the dental office manager in accordance with the data submitted to the local Public Health Department. As per national regulations, reported cases were validated by the local Public Health Department with a positive RT-PCR test. The calculated infection rate for the period between March 2020 and March 2021 was 6.37%.

An assessment of the contamination rate among dental practitioners is indicated in Figure 4. High contamination rates were registered in October 2020 (48, 20.1%), November 2020 (66, 27.7%) and December 2020 (52, 21.85%) (Figure 4a). The SARS-CoV-2 infection rate among dental practitioners is similar to the “second wave” infection rate for the general population group (Figure 4b). Reported data on infected employees working in the assessed dental offices indicates that 117 (49.2%) were doctors, 91 (38.2%) nurses, 5 (2.1%) technicians, 4 (1.7%) managers, and 21 (8.8%) auxiliary staff (Figure 5). In most of the cases, contamination occurred at home (114 cases, 47.8%) or resulted from events attendance (23 cases, 9.6%) (Figure 6). In the case of 13 (2.56%) dental offices, 21 (8.82%) “infections at work” were declared. For 61 (25.6%) infections, no information on the source of contamination was indicated.

Most of the contaminated employees of the dental offices included in our survey were without comorbidities (162, 68.1%). In the case of the 156 (30.1%) dental offices affected by infections, 32 comorbidities were declared. Estimates of the comorbidities represented in this study reveal that 19 (59.3%) were associated with obesity, alone or together with other diseases, and 6 (18.7%) were cardiopulmonary diseases. One hundred and twenty-seven (81.41%) of the 156 dental offices were private. Fifty-one (21.43%) of the total number of infected staff (238) members worked in more than one dental office.

The majority of the infected dental practitioners presented mild (145 cases, 60.9%) to moderate symptoms (57, 2.9%) (Figure 7). Our calculations show 217 (91.1%) recovered cases and two deaths, while the rest of the patients were still in the recovery stage when the survey was answered.

Collected data on rural dental offices shows that only seven practices (17.5%) reported COVID-19 cases. These results suggest an important rural-urban distinction as those cases (13 infections) account for 5.4% of the total of 238 COVID-19 declared cases.

The highest contamination rate was observed in October 2020, with five (38.4%) cases identified. Contamination occurred at home (in the family) in seven (53.8%) cases, none at work, and symptoms were mostly mild (nine, 69.2%).

Logistic regression was performed for protective measures in the dental office and risk of infection. Out of all protective measures, we found rigorous time gaps between patients (X1), staff testing (X2), quick tests in the dental office (X3), and phone triage (X4) were significantly associated with the presence of positive dental staff contaminated in dental offices. 

For overall regression *p*-value = 0.00005748. The logistic regression model using ln(odds) = b0 + b1X1 +...+bpXp (ln(odds) = 0.2159 − 0.9854 X1 + 0.5003 X2 + 0.5098 X3 − 0.5428 X4)), provided a better fit than the model without independent variables. All the independent variables (Xi) were significant.

## 4. Discussion

The present cross-sectional study was the first to assess the SARS-CoV-2 contamination rate among Romanian dental practitioners over a one-year period, with the aim of identifying the impact of the implemented prevention measures in dental settings. 

The survey was completed by 507 dentists, the majority being general dentists working in private settings from urban areas. The demographic distribution of the dental offices enrolled in the present survey was in accordance with data provided by the National Institute of Statistics in 2018. Out of 16,457 dentists, 1619 worked in the public sector and 14,034 in the private sector (The situation of health personnel in Romania—on 31 December 2018, National Institute of Public Health, National Centre for Statistics and Informatics in Public Health, March 2020). The distribution of health personnel by areas of residence was determined by the territorial distribution of health units. In 2018, the health units in the urban environment had a number of 14,426 dentists (87.7% of the total number of dentists). Concerning the number of healthcare professionals delivering health services to the population, it is important to highlight that they are underrepresented in rural areas. Thus, in 2018, for example, only 12.3% (2031) doctors [17] were carrying out their activity in rural health units, and similar results were reported in Poland by Dalewski et al. [18]. The majority of respondents were in the age groups of 30–39 years and 40–49 years, as in the studies of Ugo C et al. [19] and Chamorro-Petronacci et al. [20].

During the state of emergency declared in Romania (18 March–18 May 2020), only 39 of the 507 dental offices enrolled in this study provided emergency dental treatments. A critical shift in attendance at dental offices was therefore recorded. On the one hand, patient addressability was linked strictly with emergency treatment, and, on the other hand, teledentistry appeared to be a promising tool in the remote management of some dental cases. [21,22].

Related patient attitudes and behavior could also be found in other countries; our study supports the findings that COVID-19 significantly impacted peoples’ dental care-seeking behavior in Cluj-Napoca, Romania [16,23,24,25].

After the state of emergency was lifted in Romania and the state of alert was thereby introduced, the addressability for emergency care in the dental office was comparable to the pre-COVID-19 period, and this was most probably due to the fact that emergency treatments had also been performed during the state of emergency [16]. Activity in the assessed dental settings partially changed in 60.6% of the cases, as indicated by other authors [23].

Therefore, the two-month closure of dental offices during the state of emergency, as well as the shift in patient behavior and attitude towards dental treatments, along with the implementation of the new COVID-19 preventive protocols, are considered to have had a strong financial impact on the assessed dental offices. Similar studies in other European countries have shown a significant reduction in treatments over a period of six months following the outbreak of the COVID-19 pandemic, as well as an increase in treatment fees [18]. Our survey provides evidence that in Romania, 75.2% of the assessed dental offices’ income was reported to be lower than in 2019. Out of the assessed dental offices facing a decline in revenues, 27.25% increased dental treatment fees, and 8.5% introduced a COVID-19 tax. However, our analysis of the collected data revealed that the effect of such measures was not statistically significant.

Another noteworthy result of our survey, also confirmed by previous studies [26,27], is that more than half of the patients (57%) visiting dental offices after the outbreak of the pandemic showed concern regarding the possibility of cross-contamination in dental settings, while 55.4% inquired about the prevention measures undertaken by the dental office. Patient appointment cancellation due to SARS-CoV-2 contamination was estimated to 0%–10% in 78% of the assessed dental offices, and patient volumes were lower in 2020 than in 2019.

When assessing the implementation of national and international COVID-19 guidelines for dental practice, we found that 97.83% of the surveyed dental offices implemented protective equipment for doctors [28] (PPE), 82.05% patient triage, 89.74% proper time gaps between patients, and 95.66% patient temperature checks. In line with international literature, this study also provides evidence for the effectiveness of PPE as an essential regulation for preventing the spread of the virus to and from healthcare providers and patients [29,30,31]. Enhanced protective equipment for dental practitioners, patient phone triage, allowing regular time gaps between patients are thus considered to be more efficient than gloves, caps, and robes for patients.

A total of 3735 dental practitioners worked in the 507 dental offices included in our survey, and 1811 were doctors. Between 18 March 2020 and 1 March 2021, the contamination rate among the dental practitioners involved in our study was 6.37%. Compared to other published results [32], our study identifies a relatively higher contamination rate. This is mainly due to the fact that this rate was calculated based on data covering almost a one-year period of dental care provision during the COVID-19 pandemic. The COVID-19 contamination rate among doctors was at 6.46%. An increased number of COVID-19 cases was registered in September, October, November, and December 2020, and similar results were reported for the “second wave” infection rate in the general Romanian population. However, during the first wave of infection, the percentage of contamination among dental practitioners, compared to the total number of contaminations in Romania, was higher (2.52% in March 2020, compared to 0.26% from March 2020 to February 2021). Our hypothesis is that protective measures were not implemented at that time, and dental practitioners were highly exposed. The percentage of contaminated dental practitioners (0.42%) versus the total number of contaminations in Romania (1.25%) decreased in April; it can be assumed that this was due to the lockdown of dental offices. In May, the percentage of contaminated dentists (1.68%), compared to the percentage of COVID-19 diseased people (0.87%) in the population, increased but was lower than compared to March. This phenomenon was fostered by two main factors, the reopening of dental offices and the implementation of protective measures.

In most cases, contamination took place at home or was linked to event attendance, while contamination in the dental office was at 2.53%. The COVID-19 positive group included 32 types of comorbidities, and 59.3% were associated with obesity alone or together with other diseases, which indicates a greater susceptibility to infection among individuals with obesity [33].

It is important to acknowledge that the limited amount of data covering dental offices from rural areas was likely to have contributed to the low contamination rate observed in this study. Nevertheless, in rural areas, household transmission (53.8%) was also considered to be the primary mode of contamination.

After the onset of the vaccination program in Romania, in January 2021, as medical staff were given priority, a decrease in infection rate among the assessed dental offices was registered. These results are in line with reported data on general medical staff [34].

The limitations of our study relate to the fact that the survey was distributed via the internet (e-mail or social media) on different dentistry forums and discussion rooms. However, the strength of this study lies in the fact that it provides relevant data about the COVID-19 contamination rate among Romanian dental practitioners over a one-year period and includes detailed information on the implementation of preventive measures in dental offices.

## 5. Conclusions

Most assessed dental offices have implemented the latest COVID-19 prevention guidelines. The full reopening was implemented in May 2020 in all assessed Romanian dental offices, and elective dental procedures were allowed to be performed again. The decline in patient volume and the implementation of preventive measures led to financial difficulties for the majority of the assessed dental offices. Over the course of one year, during which dental offices functioned in accordance with the pandemic response measures, the contamination rate among dental practitioners was 6.37%. As most contamination occurred outside the dental office, this study underlines an overall greater level of safety and an increased patient addressability in dental offices.

## Figures and Tables

**Figure 1 ijerph-18-04897-f001:**
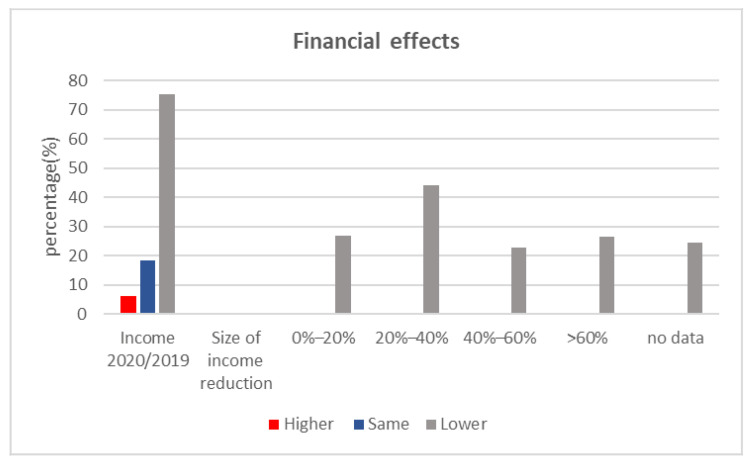
Financial effects generated by the COVID-19 pandemic in dental offices.

**Figure 2 ijerph-18-04897-f002:**
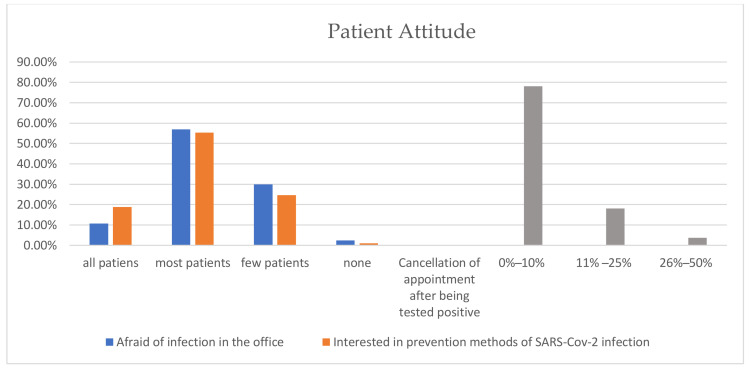
Patient attitude towards dental treatment during COVID-19 pandemic.

**Figure 3 ijerph-18-04897-f003:**
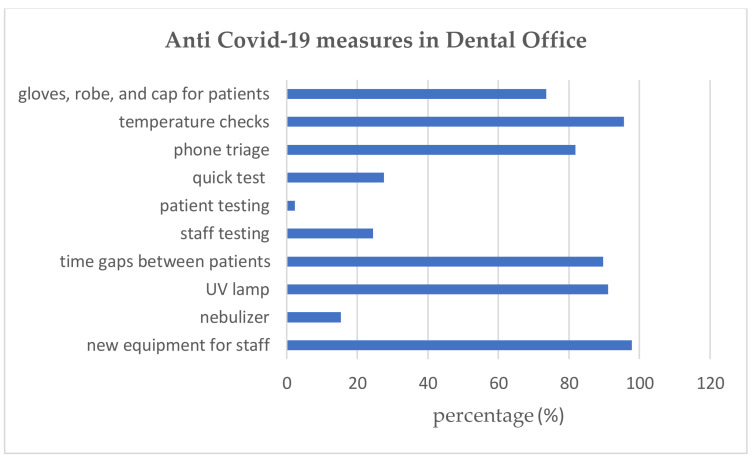
Protective measures in dental offices.

**Figure 4 ijerph-18-04897-f004:**
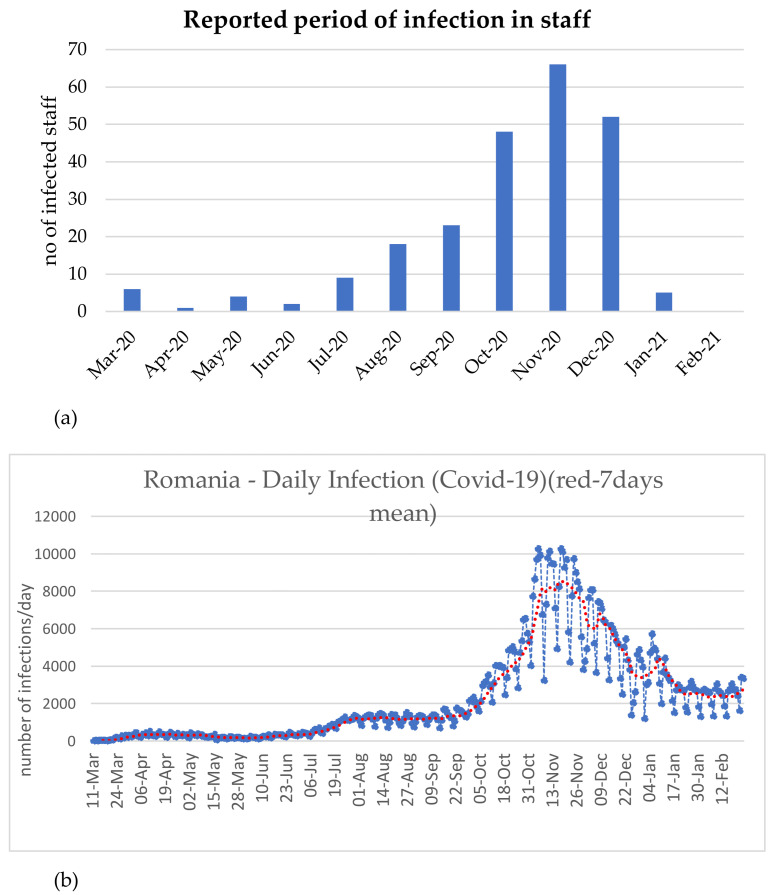
(**a**) Contamination rate among dental practitioners in the dental office. (**b**) General Romanian population infection rate.

**Figure 5 ijerph-18-04897-f005:**
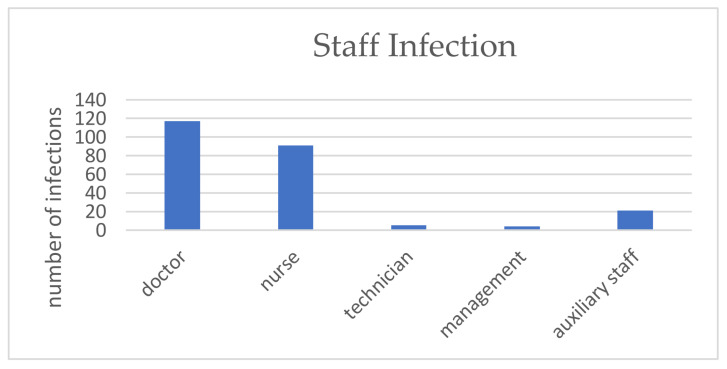
COVID-19 positive staff category.

**Figure 6 ijerph-18-04897-f006:**
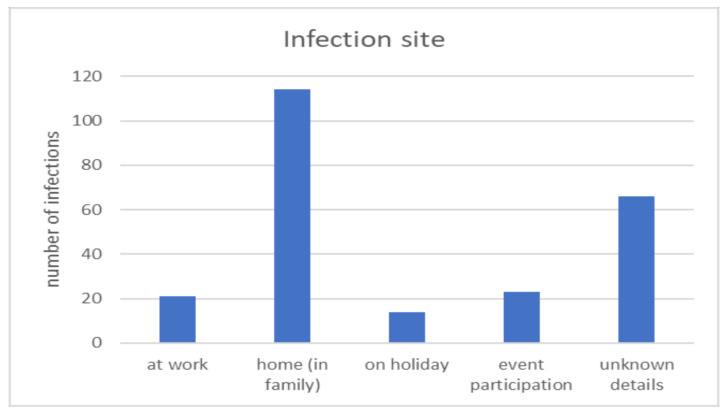
Source of contamination.

**Figure 7 ijerph-18-04897-f007:**
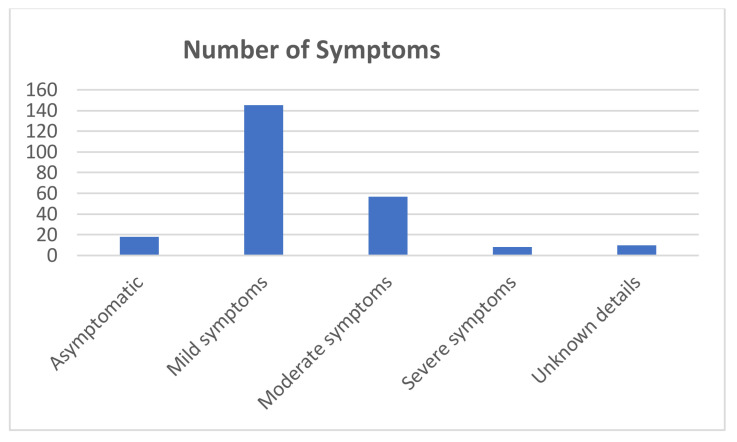
Symptoms of the infected dental practitioners.

**Table 1 ijerph-18-04897-t001:** Economic changes generated by the COVID-19 pandemic in dental offices.

Activity in Dental Offices			
Dental offices providing dental services during the state of emergency period	Yes	No	
	39 (7.7%)	468 (92.3%)	
Number of emergency treatments after lockdown	Higher	Lower	Same 270 (53.3%)
	89 (17.6%)	148 (29.92%)	
Activity in the dental office	Major changes	Partial changes	No changes 15 (3%)
	185 (36.5%)	307 (60.6%)	
Demand for elective dental services	Higher	Lower	Same 272(53.6%)
	22 (4.3%)	213 (42%)	
Effect on the Team Size			
Team size	Reduced team	Enlarged team	Same size 435 (85.8%)
	50 (9.9%)	22 (4.3%)	

**Table 2 ijerph-18-04897-t002:** Patient volume in 2020 compared to 2019.

	20-May	20-June	20-July	20-August	20-September	20-October	20-November	20-December	21-January	21-February
	No/%	No/%	No/%	No/%	No/%	No/%	No/%	No/%	No/%	No/%
lower	367	300	269	278	270	259	276	302	180	128
	72.39%	59.17%	53.06%	54.83%	53.25%	51.08%	54.44%	59.57%	35.50%	25.25%
higher	42	64	66	56	56	61	65	56	28	16
	8.28%	12.62%	13.02%	11.05%	11.05%	12.03%	12.82%	11.05%	5.52%	3.16%
same	31	96	124	120	134	138	119	104	56	42
	6.11%	18.93%	24.40%	23.67%	26.43%	27.22%	23.47%	20.51%	11.05%	8.28%
no data/no response	67	47	48	53	47	49	47	45	243	321
	13.02%	9.27%	9.47%	10.45	9.47%	9.67%	9.27%	8.88%	47.93%	63.31%

## Data Availability

The datasets used and analyzed during the current study are available from the corresponding author on reasonable request.

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
