# Peer review of "Assessing SARS-CoV-2 Infection Rate among Romanian Dental Practitioners"

_ijerph, 2021, doi:10.3390/ijerph18094897_

Round 1

Reviewer 1 Report

Manuscript ID: applsci-901016

Assessing SARS-CoV-2 infection rate among Romanian dental practitioners

While the overall idea for this engineering and research effort is good, the presentation of it in the paper minor substantial improvement.

In this study, the authors explain the infection rate among dental practitioners from Romania and analyze the economic impact of Covid-19 on dental offices.

At the same time, the survey included questions that assessed demographic data of the dentists who completed the questionnaire, economic aspects and, epidemiological aspects related to COVID-19 among dental practitioners.

This manuscript is well written. I recommend publication of this manuscript without any changes.

Author Response

Dear Editors,

            Thank you for allowing us to resubmit a revised version of our manuscript. Please accept our revised version for further consideration. We would like to express our gratitude to the reviewers for providing constructive feedback by identifying the areas of our manuscript that needed further improvements. We appreciate the tremendous effort and time the reviewers devoted to strengthening our manuscript. Accordingly, we have uploaded the revised manuscript with all the changes indicated with red. Please find below our response to each of the reviewers’ comments. We hope this will ease the reading of the paper and we are confident that the new version of the manuscript is significantly improved. Thus, we look forward to hearing from you and to respond to any other questions or comments you may have. As the audience of your prestigious journal deserves discussion on the topic of COVID-19, we are submitting the revised version of the manuscript and we will appreciate your quick response.

With my best regards,

Antonia Boca

(on behalf of all coauthors)

Reviewer 1

Open Review

English language and style

( ) Extensive editing of English language and style required
( ) Moderate English changes required
(x) English language and style are fine/minor spell check required
( ) I don't feel qualified to judge about the English language and style

Yes

Can be improved

Must be improved

Not applicable

Does the introduction provide sufficient background and include all relevant references?

( )

(x)

( )

( )

Is the research design appropriate?

( )

(x)

( )

( )

Are the methods adequately described?

( )

(x)

( )

( )

Are the results clearly presented?

( )

(x)

( )

( )

Are the conclusions supported by the results?

( )

(x)

( )

( )

Comments and Suggestions for Authors

Assessing SARS-CoV-2 infection rate among Romanian dental practitioners

While the overall idea for this engineering and research effort is good, the presentation of it in the paper minor substantial improvement.

In this study, the authors explain the infection rate among dental practitioners from Romania and analyze the economic impact of Covid-19 on dental offices.

At the same time, the survey included questions that assessed demographic data of the dentists who completed the questionnaire, economic aspects and, epidemiological aspects related to COVID-19 among dental practitioners.

This manuscript is well written. I recommend publication of this manuscript without any changes.

Thank you very much for the time you devoted to this manuscript, and for appreciating our work.

Reviewer 2 Report

The authors have put enormous effort and time into this study and the presentations of their findings. However, there are some issues that need to be revised before the article is acceptable for publication.

How did the authors choose the participants of the survey? Did they use discussion groups or dental chamber email base ?

The survey was completed by only one dental practitioner from each dental clinic- why and how was it decided which one? Was it medical director  or maybe the oldest,  or the owner? These need clarification.

A total of 3,735 dental practitioners were working in the 507 assessed dental offices, 197 and among them 238 COVID-19 cases of contamination were recorded, of which: 61 198 (25.63%) males, 177 (74.37%) females, and the most affected ages being 30-39 (97, 40.76%) 199 and 40-49 (68, 28.57%). The calculated infection rate for the period between March 2020 200 and March 2021 was 6.37% – how was the infection confirmed, what kind of tests were used?

Regarding statistical analysis. I’d strongly suggest performing logistic regression model for protective measures in dental offices and risk of infection.

Also, it would be beneficial if you included some other works on Sars-Cov2 infection, at least in Europe, as there are numbers of publications on this topic available. I feel that incorporating more references into discussion section would greatly improve the overall quality of the study. Some factors mentioned in your article may be cross-linked with findings from other European countries, e.g.

Dalewski, B.; Palka, L.; Kiczmer, P.; Sobolewska, E. The Impact of SARS-CoV-2 Outbreak on the Polish Dental Community’s Standards of Care—A Six-Month Retrospective Survey-Based Study. Int. J. Environ. Res. Public Health 202118, 1281. https://doi.org/10.3390/ijerph18031281

Bizzoca, M.E.; Campisi, G.; Lo Muzio, L. Covid-19 Pandemic: What Changes for Dentists and Oral Medicine Experts? A Narrative Review and Novel Approaches to Infection Containment. Int. J. Environ. Res. Public Health 202017, 3793. https://doi.org/10.3390/ijerph17113793

Chamorro-Petronacci, C.; Martin Carreras-Presas, C.; Sanz-Marchena, A.; A Rodríguez-Fernández, M.; María Suárez-Quintanilla, J.; Rivas-Mundiña, B.; Suárez-Quintanilla, J.; Pérez-Sayáns, M. Assessment of the Economic and Health-Care Impact of COVID-19 (SARS-CoV-2) on Public and Private Dental Surgeries in Spain: A Pilot Study. Int. J. Environ. Res. Public Health 202017, 5139. https://doi.org/10.3390/ijerph17145139

Please also paraphrase lines 255-257 as they are directly taken from the article “Impact of COVID-19 on Dental Emergency Services in Cluj-Napoca Metropolitan Area: A Cross-Sectional Study”. Additionally, some language editing is recommended as there are some minor mistakes and unnecessary repetitions, which make reading difficult.

Author Response

Dear Editors,

            Thank you for allowing us to resubmit a revised version of our manuscript. Please accept our revised version for further consideration. We would like to express our gratitude to the reviewers for providing constructive feedback by identifying the areas of our manuscript that needed further improvements. We appreciate the tremendous effort and time the reviewers devoted to strengthening our manuscript. Accordingly, we have uploaded the revised manuscript with all the changes indicated with red. Please find below our response to each of the reviewers’ comments. We hope this will ease the reading of the paper and we are confident that the new version of the manuscript is significantly improved. Thus, we look forward to hearing from you and to respond to any other questions or comments you may have. As the audience of your prestigious journal deserves discussion on the topic of COVID-19, we are submitting the revised version of the manuscript and we will appreciate your quick response.

With my best regards,

Antonia Boca

(on behalf of all coauthors)

Reviewer 2

Open Review

English language and style

( ) Extensive editing of English language and style required
(x) Moderate English changes required
( ) English language and style are fine/minor spell check required
( ) I don't feel qualified to judge about the English language and style

Yes

Can be improved

Must be improved

Not applicable

Does the introduction provide sufficient background and include all relevant references?

(x)

( )

( )

( )

Is the research design appropriate?

( )

(x)

( )

( )

Are the methods adequately described?

( )

(x)

( )

( )

Are the results clearly presented?

( )

( )

(x)

( )

Are the conclusions supported by the results?

(x)

( )

( )

( )

Comments and Suggestions for Authors

The authors have put enormous effort and time into this study and the presentations of their findings. However, there are some issues that need to be revised before the article is acceptable for publication.

  1. How did the authors choose the participants of the survey? Did they use discussion groups or dental chamber email base?

Thank you for your suggestion, we added the required information in the manuscript.

The survey was distributed via the internet (e-mail or social media) on different dentistry forums and discussion rooms.

  1. The survey was completed by only one dental practitioner from each dental clinic- why and how was it decided which one?  - Was it medical director or maybe the oldest, or the owner? These need clarification.

Thank you for your suggestion. In section 1 of the questionnaire we specified that -Only one member of a clinic / practice is asked to complete this questionnaire to avoid double reporting. Each dental office chose the person to complete the questionnaire. We added in the manuscript:

338 (66.70%) of the persons who completed the questionnaire were managers and the rest 169 (33.34%) employees.

  1. A total of 3,735 dental practitioners were working in the 507 assessed dental offices, 197 and among them 238 COVID-19 cases of contamination were recorded, of which: 61 198 (25.63%) males, 177 (74.37%) females, and the most affected ages being 30-39 (97, 40.76%) 199 and 40-49 (68, 28.57%). The calculated infection rate for the period between March 2020 200 and March 2021 was 6.37% – how was the infection confirmed, what kind of tests were used?

Thank you for your suggestion, we added this information in the manuscript.

All these cases were reported by the dental office manager in accordance to the data submitted to the local Public Health Department. As per national regulations, reported cases were validated by the local Public Health Department with a positive RT-PCR test.

  1. Regarding statistical analysis. I’d strongly suggest performing logistic regression model for protective measures in dental offices and risk of infection.

Thank you very much for providing constructive feedback. As suggested by the reviewer we performed logistic regression model for protective measures and risk of infection. Only the protective measures listed in the questionnaire were included in the analyses and correlated with the presence of contamination among dental practitioners. We added the information in the Material and Method and Result section.

Logistic regression model was performed using AAT Bioquest, Inc. (2021, April 27) Quest Graph™ Logistic Regression (Logit) Calculator.". Retrieved from https://www.aatbio.com/tools/logistic-regression-logit-calculator.

Logistic regression model was performed for protective measures in the dental office and risk of infection. Out of all protective measures, we found that: rigorous time gaps between patients (X1), staff testing (X2), quick test in the dental office (X3) and phone triage (X4) are statistic significantly associated with the presence of positive dental staff contaminated in the dental office.

For overall regression p-value = 0.00005748. The logistic regression model with ln(odds) = b0+ b1X1 +...+bpXp (ln(odds) = 0.2159 - 0.9854 X1 + 0.5003 X2 + 0.5098 X3 - 0.5428 X4)), provides a better fit than the model without the independent variables. All the independent variables (Xi) are significant.

  1. Also, it would be beneficial if you included some other works on Sars-Cov2 infection, at least in Europe, as there are numbers of publications on this topic available. I feel that incorporating more references into discussion section would greatly improve the overall quality of the study. Some factors mentioned in your article may be cross-linked with findings from other European countries, e.g.

  • Dalewski, B.; Palka, L.; Kiczmer, P.; Sobolewska, E. The Impact of SARS-CoV-2 Outbreak on the Polish Dental Community’s Standards of Care—A Six-Month Retrospective Survey-Based Study. Int. J. Environ. Res. Public Health 2021, 18, 1281. https://doi.org/10.3390/ijerph18031281
  • Bizzoca, M.E.; Campisi, G.; Lo Muzio, L. Covid-19 Pandemic: What Changes for Dentists and Oral Medicine Experts? A Narrative Review and Novel Approaches to Infection Containment. Int. J. Environ. Res. Public Health 2020, 17, 3793. https://doi.org/10.3390/ijerph17113793
  • Chamorro-Petronacci, C.; Martin Carreras-Presas, C.; Sanz-Marchena, A.; A Rodríguez-Fernández, M.; María Suárez-Quintanilla, J.; Rivas-Mundiña, B.; Suárez-Quintanilla, J.; Pérez-Sayáns, M. Assessment of the Economic and Health-Care Impact of COVID-19 (SARS-CoV-2) on Public and Private Dental Surgeries in Spain: A Pilot Study. Int. J. Environ. Res. Public Health 2020, 17, 5139. https://doi.org/10.3390/ijerph17145139

Thank you for your suggestion, we indeed found the sources you suggested important for our study and we incorporated them as it follows in the discussion sector.

Concerning the number of healthcare professionals delivering health services to the population, it is important to highlight that they are underrepresented in rural areas. Thus, in 2018, for example, only 12.3% (2031) doctors [17] were carrying out their activity in rural health units, and similar results were reported in Poland by Dalewski et all [18]. The majority of respondents were from the age group 30-39 years, and 40-49 years, as in the studies of Ugo C et al [19] and Chamorro-Petronacci et al.[20]

Similar studies in other European countries have shown a significant reduction in treatments during a period of 6 months following the outbreak of the COVID-19 pandemic, as well as an increase in treatment fees.[18]

When assessing the implementation of national and international COVID-19 guidelines for dental practice, we found that 97.83% of the surveyed dental offices implemented protective equipment for doctors [28] (PPE), 82.05% patient triage, 89.74% proper time gaps between patients and 95.66% patient temperature checks. In line with international literature, this study also provides evidence of effectiveness of PPE as an essential regulation for preventing the spread of the virus to and from health-care pro-viders and patients [29- 31]

  1. Dalewski, B.; Palka, L.; Kiczmer, P.; Sobolewska, E. The Impact of SARS-CoV-2 Outbreak on the Polish Dental Community’s Standards of Care—A Six-Month Retrospective Survey-Based Study. J. Environ. Res. Public. Health 2021, 18, 1281.
  2. Chamorro-Petronacci, C.; Martin Carreras-Presas, C.; Sanz-Marchena, A.; A Rodríguez-Fernández, M.; María Suárez-Quintanilla, J.; Rivas-Mundiña, B.; Suárez-Quintanilla, J.; Pérez-Sayáns, M. Assessment of the Economic and Health-Care Impact of COVID-19 (SARS-CoV-2) on Public and Private Dental Surgeries in Spain: A Pilot Study. J. Environ. Res. Public. Health 2020, 17, 5139.
  3. Bizzoca, M.E.; Campisi, G.; Lo Muzio, L. Covid-19 Pandemic: What Changes for Dentists and Oral Medicine Experts? A Narrative Review and Novel Approaches to Infection Containment. J. Environ. Res. Public. Health 2020, 17, 3793.

  1. Please also paraphrase lines 255-257 as they are directly taken from the article “Impact of COVID-19 on Dental Emergency Services in Cluj-Napoca Metropolitan Area: A Cross-Sectional Study”. Additionally, some language editing is recommended as there are some minor mistakes and unnecessary repetitions, which make reading difficult.

Thank you for your suggestion, as recommended by the reviewer we paraphrased lines 255-257.

A critical shift in attendance at dental offices was therefore recorded. On the one hand, patient addressability was linked strictly with emergency treatment and, on the other hand, teledentistry appeared to be a promising tool in the remote management of some dental cases. [21, 22]

  1. Giudice, A.; Barone, S.; Muraca, D.; Averta, F.; Diodati, F.; Antonelli, A.; Fortunato, L. Can Teledentistry Improve the Moni-toring of Patients during the Covid-19 Dissemination? A Descriptive Pilot Study. Int J Environ Res Public Health 2020, 17, (10): 3399.
  2. Bennardo, F.; Antonelli, A.; Barone, S.; Figliuzzi, M.M.; Fortunato, L.; Giudice, A. Change of Outpatient Oral Surgery during the COVID-19 Pandemic: Experience of an Italian Center. Int J Dent 2020, 2020, 8893423.

Reviewer 3 Report

The manuscript submitted on IJERPH titled "Assessing SARS-CoV-2 infection rate among Romanian dental practitioners" is a cross-sectional study conducted using a web-based survey on the infection rate among dental practitioners from Romania and to analyse the economic impact of COVID-19 on dental offices.

The manuscript is another paper talking about the spread of COVID-19 in a precise district of Europe (Romania), however can add other information about the situation of COVID-19 among dental practitioners.

I suggest making some changes in the text:

- Edit the abstract to make it more attractive to readers. Avoid a list of data by giving some more food for thought on the aim of the manuscript

- Introduction:

"In dental offices, aerosol-generating dental procedures (AGDPs) are amongst the most common types of treatment, thus concerns about the risk of transmission from patient-to-practitioner and patient-to-patient arose [5,6]"..

I suggest to add some interesting reference about the risk of transmission, also in other countries 
DOI: 10.2174/1874210602014010298
doi: 10.1631/jzus.B2010010

- Methods are good expressed

- Results

Please, summarizes the results, using not only tables but also graphs

- Discussion

I am in accordance to the infection rate report among the dental practitioners, in particular during the second wave of infection.
I suggest to the authors to improve this part discussing possible hypothesis and difference with the first wave of infection.

"As routine dental care was not available during the state of emergency, and the number of dental care providers was smaller, patients were only seeking emergency treatments, marking a critical shift in attendance to dental services"

I suggest to improve these sentences referring to the emergency dental services and on the change of therapies given to patients, highlighting the importance of telemedicine.

  • PMID: 32733566
  • doi: 10.3390/ijerph17103399

- Conclusion
Are in accordance to the results and discussion part

Although the article has limited relevance in the scenario of the spread of covid-19 infection, giving additional information on the conditions present among Romanian dentists, the article can be evaluated for publication after the changes suggested by the reviewers

Author Response

Dear Editors,

            Thank you for allowing us to resubmit a revised version of our manuscript. Please accept our revised version for further consideration. We would like to express our gratitude to the reviewers for providing constructive feedback by identifying the areas of our manuscript that needed further improvements. We appreciate the tremendous effort and time the reviewers devoted to strengthening our manuscript. Accordingly, we have uploaded the revised manuscript with all the changes indicated with red. Please find below our response to each of the reviewers’ comments. We hope this will ease the reading of the paper and we are confident that the new version of the manuscript is significantly improved. Thus, we look forward to hearing from you and to respond to any other questions or comments you may have. As the audience of your prestigious journal deserves discussion on the topic of COVID-19, we are submitting the revised version of the manuscript and we will appreciate your quick response.

With my best regards,

Antonia Boca

(on behalf of all coauthors)

Reviewer 3

Open Review

English language and style

( ) Extensive editing of English language and style required
( ) Moderate English changes required
( ) English language and style are fine/minor spell check required
(x) I don't feel qualified to judge about the English language and style

Yes

Can be improved

Must be improved

Not applicable

Does the introduction provide sufficient background and include all relevant references?

( )

( )

(x)

( )

Is the research design appropriate?

(x)

( )

( )

( )

Are the methods adequately described?

( )

(x)

( )

( )

Are the results clearly presented?

(x)

( )

( )

( )

Are the conclusions supported by the results?

(x)

( )

( )

( )

Comments and Suggestions for Authors

The manuscript submitted on IJERPH titled "Assessing SARS-CoV-2 infection rate among Romanian dental practitioners" is a cross-sectional study conducted using a web-based survey on the infection rate among dental practitioners from Romania and to analyse the economic impact of COVID-19 on dental offices.

The manuscript is another paper talking about the spread of COVID-19 in a precise district of Europe (Romania), however can add other information about the situation of COVID-19 among dental practitioners.

I suggest making some changes in the text:

  1. Edit the abstract to make it more attractive to readers. Avoid a list of data by giving some more food for thought on the aim of the manuscript

Thank you for your constructive feedback. We have made some changes in the abstract.

Due to the impact of Covid-19 pandemic on dental treatments, the present research aimed to assess the infection rate among dental practitioners from Romania and to analyse the economic impact of Covid-19 on dental offices. We designed a web-based survey distributed to dental practitioners from Romania. The survey included questions that assessed demographic data from the dentists who completed the questionnaire, along with economic aspects and, epidemiological aspects related to the impact of COVID-19 pandemic on dental practitioners. 507 dentists completed the survey. Three quarters 382 (75.2%) of the assessed dental offices reported a decrease in the income and patient volume compared to 2019. More than half of the pPatients visiting the dental office paid more attention to the risk of infection (289 (57%) cases), and to prevention methods (281 (55.4%)). Most dental offices, 496 (97.83%), implemented specific protective equipment for doctors. 3735 dental practitioners were working in the 507 assessed dental offices, and among them 238 COVID-19 cases of contamination were recorded. High contamination rates were registered in October (48, 20.1%), November (66, 27.7%) and December 2020 (52, 21.85%). Contamination mostly took place at home (114 cases, 47.8%) or resulted from event attendance (23 cases, 9.6%). This study underlines an overall greater level of safety and an increased patient addressability in dental offices

  1. Introduction:

"In dental offices, aerosol-generating dental procedures (AGDPs) are amongst the most common types of treatment, thus concerns about the risk of transmission from patient-to-practitioner and patient-to-patient arose [5,6]"..

I suggest to add some interesting reference about the risk of transmission, also in other countries .

DOI: 10.2174/1874210602014010298

doi: 10.1631/jzus.B2010010

Thank you for your suggestion, as requested by the reviewer we added other interesting references about the risk SARS-CoV-2 transmission in the dental office. [7,8]

  1. Giudice, A.; Bennardo, F.; Antonelli, A.; Barone, S.; Fortunato, L. COVID-19 Is a New Challenge for Dental Practitioners: Advice on Patients’ Management from Prevention of Cross Infections to Telemedicine. Open Dent. J. 2020, 14, doi:10.2174/1874210602014010298.
  2. Ge, Z.-Y.; Yang, L.-M.; Xia, J.-J.; Fu, X.-H.; Zhang, Y.-Z. Possible Aerosol Transmission of COVID-19 and Special Precautions in Dentistry. J. Zhejiang Univ. Sci. B 2020, 21, 361–368, doi:10.1631/jzus.B2010010.

Methods are good expressed

Thank you very much for appreciating our work.

  1. Results

Please, summarizes the results, using not only tables but also graphs

Thank you very much for your suggestion. As requested by the reviewer, we summarized our results using not only tables but also graphs.

We deleted the financial effects from Table 1 and added this aspect in Figure 1. Financial Effects generated by the COVID-19 pandemic in dental offices

We deleted Table 2. Patient attitude towards dental treatment during COVID-19 pandemic, and added this information in Figure 2. Patient attitude towards dental treatment during COVID-19 pandemic.

Table 3 Patient volume in 2020 compared to 2019, became Table 2.

We deleted Table 4. Protective measures in dental offices and added Figure 3: Protective measures in dental offices.

Figure 1 became Figure 4.

Figure 2 became Figure 5.

We added Figure 6. Source of contamination.

We added Figure 7. Symptoms of the infected dental practitioners

  1. Discussion

I am in accordance to the infection rate report among the dental practitioners, in particular during the second wave of infection.

I suggest to the authors to improve this part discussing possible hypothesis and difference with the first wave of infection.

Thank you very much for your suggestion. As requested by the reviewer, we discussed possible hypothesis and difference with the first wave of infection.

However, during the first wave of infection, the percentage of contamination among dental practitioners, compared to the total number of contaminations in Romania, was higher (2.52% - in March 2020, compared to 0.26% - from March 2020 to February 2021). Our hypothesis is that protective measures were not implemented at that time and dental practitioners were highly exposed. The percentage of contaminated dental practitioners (0.42%) versus total number of contaminations in Romania (1.25%) decreased in April, it can be assumed that this is due to the lockdown of dental offices. In May, the percentage of contaminated dentists (1.68%), compared to the percentage of Covid-19 diseased people (0.87%) in the population, increased, but was lower compared to March. This phenomenon was fostered by two main factors, the reopening of dental offices and the implementation of protective measures.

  1. "As routine dental care was not available during the state of emergency, and the number of dental care providers was smaller, patients were only seeking emergency treatments, marking a critical shift in attendance to dental services"

I suggest to improve these sentences referring to the emergency dental services and on the change of therapies given to patients, highlighting the importance of telemedicine.

  • PMID: 32733566
  • doi: 10.3390/ijerph17103399

Thank you very much for your suggestion, as suggested by the reviewer we introduced the idea of telemedicine in time of Covid-19 and we added 2 references.

A critical shift in attendance at dental offices was therefore recorded. On the one hand, patient addressability was linked strictly with emergency treatment and, on the other hand, teledentistry appeared to be a promising tool in the remote management of some dental cases. [21, 22]

  1. Giudice, A.; Barone, S.; Muraca, D.; Averta, F.; Diodati, F.; Antonelli, A.; Fortunato, L. Can Teledentistry Improve the Moni-toring of Patients during the Covid-19 Dissemination? A Descriptive Pilot Study. Int J Environ Res Public Health 2020, 17, (10): 3399.
  2. Bennardo, F.; Antonelli, A.; Barone, S.; Figliuzzi, M.M.; Fortunato, L.; Giudice, A. Change of Outpatient Oral Surgery during the COVID-19 Pandemic: Experience of an Italian Center. Int J Dent 2020, 2020, 8893423.

 Conclusion

Are in accordance to the results and discussion part

Thank you very much for appreciating our work.

Although the article has limited relevance in the scenario of the spread of covid-19 infection, giving additional information on the conditions present among Romanian dentists, the article can be evaluated for publication after the changes suggested by the reviewers.

Round 2

Reviewer 3 Report

Authors followed the suggestions improving the manuscript .